# A Shallow Feature Extraction Network With A Large Receptive Field For Stereo Matching Tasks

## Abstract

Stereo matching is one of the important basic tasks in the computer vision field. In recent years, stereo matching algorithms based on deep learning have achieved excellent performance and become the mainstream research direction. Existing algorithms generally use deep convolutional neural networks (DCNNs) to extract more abstract semantic information, but we believe that the detailed information of the spatial structure is more important for stereo matching tasks. Based on this point of view, this paper proposes a shallow feature extraction network with a large receptive field. The network consists of three parts: a primary feature extraction module, an atrous spatial pyramid pooling (ASPP) module and a feature fusion module. The primary feature extraction network contains only three convolution layers. This network utilizes the basic feature extraction ability of the shallow network to extract and retain the detailed information of the spatial structure. In this paper, the dilated convolution and atrous spatial pyramid pooling (ASPP) module are introduced to increase the size of receptive field. In addition, a feature fusion module is designed, which integrates the feature maps with multiscale receptive fields and mutually complements the feature information of different scales. We replaced the feature extraction part of the existing stereo matching algorithms with our shallow feature extraction network, and achieved state-of-the-art performance on the KITTI 2015 dataset. Compared with the reference network, the number of parameters is reduced by 42%, and the matching accuracy is improved by 1.9%.

## 1 Introduction

Since the introduction of deep learning in the computer vision field, increasing the network depth (that is, the number of layers in the network) seems to be a necessary means to improve the feature extraction ability. Taking the object classification task as an example, as the network depth increases from the 8-layer network AlexNet (Krizhevsky et al., 2012) to the 16-layer network VGG (Simonyan & Zisserman, 2014) and to the 101-layer network ResNet (He et al., 2015), the classification accuracy constantly improves. There are two purposes of the deep network. First, the deep network can improve the ability to extract abstract features (Zeiler & Fergus, 2013), which are important for some vision tasks, such as object detection (Girshick, 2015; Ren et al., 2017) and classification. For example, for objects such as cups, their colors, shapes and sizes may be different, and they cannot be accurately identified using only these primary feature information. Therefore, the feature extraction network must have the ability to extract more abstract semantic information. Second, the deep feature extraction network can obtain a larger receptive field to learn more context information (Luo et al., 2017; Liu et al., 2018). With the increase in the number of network layers, the size of the receptive field is also constantly increasing. In particular, after image sampling using a pooling operation, even the 3*3 convolution kernel has the ability to extract context information. Many studies (Zeiler & Fergus, 2013; Yu & Koltun, 2016) have shown that the lower part of the convolution neural network mainly extracts primary features, such as the edges and corners, while the higher part can extract more abstract semantic information. However, many basic vision tasks rely more on basic feature information instead of the high-level abstract features.

Stereo matching is one of the basic vision tasks. In the traditional stereo matching algorithm (Scharstein & Szeliski, 2002), the color similarity metrics of pixels are usually used to

calculate the matching costs between the left and right images to find the matching points in the two images. After the introduction of deep learning, more robust feature information can be obtained through training and learning, which can effectively improve the performance of the stereo matching algorithm. At present, many excellent stereo matching algorithms based on deep learning, such as the GC-Net (Kendall et al., 2017), PSMNet (Chang & Chen, 2018) and GwcNet (Guo et al., 2019), generally adopt similar processes, including feature extraction, matching cost volume construction, 3D convolution and disparity regression. This paper focuses on the feature extraction steps.

The stereo matching task has two requirements for the feature extraction network. The first requirement is the enlargement of the receptive field as far as possible so that the network can obtain more context information, which is critical to solving the mismatching problems in the discontinuous disparity area. Because a larger receptive field can learn the relationships between different objects, even if there are problems, such as conclusion or inconsistent illumination, the network can use the context information to infer disparity and improve the stereo matching accuracy in the ill-posed regions. The second requirement is the maintenance of more details of the spatial structure, which can improve the matching accuracy of many small structures, such as railings, chains, traffic signs and so on. The existing feature extraction networks usually use a deep convolution neural network to obtain a larger receptive field and extract more abstract semantic information. In this process, with the increase of the network layers and the compression of the image size, substantial detailed information of the spatial structure is inevitably lost. We believe that compared with the abstract semantic information that is extracted by a deep network, the detailed information of the spatial structure is more important to improving the stereo matching accuracy. Based on this point of view, this paper proposes a novel structure of feature extraction network – a shallow feature extraction network. Unlike the common feature extraction network (with ResNet-50 as the backbone), in this paper, the backbone of the feature extraction network only has 3 convolution layers, and the image is only downsampled once in the first convolution layer to compress the size of the image. This structure retains more details of the spatial structure and pays more attention to primary features such as the edges and corners of objects, while abandoning more abstract semantic information. To solve the problem that the size of the receptive field of the shallow structure is limited, this paper introduces the atrous spatial pyramid pooling (ASPP) module (Chen et al., 2018). The ASPP module uses the dilated convolution to increase the receptive field size without increasing the number of parameters. In addition, the convolution layers with different dilation rate can obtain feature maps with multiscale receptive fields. The large receptive fields can be used to obtain context information and to solve the problem of mismatching in ill-posed regions, and the small receptive fields can be used to retain more detailed information of the spatial structure and to improve the stereo matching accuracy in local areas. To integrate feature maps with multiscale receptive fields, this paper designs the feature fusion module and introduces the channel attention mechanism (Jie et al., 2017). We assign different weights to feature maps with different dilation rates in the channel dimensions. The weights are acquired through learning, and more weight and attention are given to the feature channels with greater roles.

The advantages of a shallow feature extraction network with a large receptive field are twofold. One advantage is that the network can meet the two requirements of the stereo matching task for the feature extraction network. On the basis of ensuring the large receptive field, more details of the spatial structure are retained. The other advantage is that the network greatly reduces the number of parameters and the difficulties of network training and deployment. The feature extraction network that is designed in this paper is used to replace the feature extraction part of the existing stereo matching network, and state-of-the-art performance is achieved on the KITTI2015 dataset (Geiger, 2012). Compared with the reference network, the number of parameters is reduced by 42%, and the matching accuracy is improved by 1.9%. The main contributions of this paper are as follows.

- A shallow feature extraction network is proposed to extract and retain more details of the spatial structure. This network can improve the stereo matching accuracy with fewer parameters.

- The dilated convolution and ASPP module are introduced to enlarge the receptive field. We verify the effect of the dilated convolution on the receptive field using mathematics and experiments.

- A feature fusion module, which integrates the feature maps with multiscale receptive fields, is designed and realizes the mutual complementary feature information of different scales.

## 2 RELATED WORK

In recent years, deep learning methods have gradually replaced traditional algorithms and become the mainstream stereo matching algorithm. The GC-Net (Kendall et al., 2017) designed a new stereo matching algorithm process based on deep learning, including feature extraction, matching cost volume construction, 3D convolution and disparity regression. First, in the feature extraction part, two deep convolution neural network with shared weights are used to extract the feature information from the left and right images. The matching cost volume is formed by cascading the left and right feature maps. Then, the 3D convolution is carried out on the matching cost volume, which can extract the feature representations from the three dimensions of height, width and disparity. Finally, the regression method is used to obtain the disparity map.

Since the introduction of GC-Net, most stereo matching algorithms follow the stereo matching process of GC-Net. Focused on the feature extraction part, this section introduces many improved schemes for feature extraction networks using excellent algorithms in recent years. The PSM-Net (Chang & Chen, 2018) further deepened the feature extraction network structure, which took ResNet-50 as the backbone, and used the spatial pyramid pooling (SPP) module (Kaiming et al., 2014) to obtain the feature information at different scales. GwcNet (Guo et al., 2019) retained the backbone structure in the PSMNet, but it eliminated the SPP module, and proposed a new method to form the matching cost volume using the group-wise correlation. Based on the PSMNet, MCUA (Nie et al., 2019) introduced DenseNet's (Huang et al., 2016) densely connected structure, which summarizes the output of each layer and transmits it to the next layer. This structure forms a dense connection between the different layers of the network. The Stereo-DRNet (Chabra et al., 2019) introduced the vortex pooling structure (Xie et al., 2018), which is a variant of the ASPP. In this structure, average pooling is carried out on the feature map before the dilated convolution, and the size of the pooling kernel is the corresponding dilation rate. Zhu et al. proposed the CFP-Net (Zhu et al., 2019) and designed a cross form spatial pyramid pooling (CFSPP) module, which consist of two branches: one branch is the SPP structure, and the other branch is the ASPP structure. The feature maps obtained from two branches are concatenated to obtain the feature information of each scale.

In the feature extraction network, almost all the existing stereo matching networks take the ResNet-50 structure as their backbones. In this paper, we proposed a shallow feature extraction network with fewer parameters but a larger receptive field, whose matching accuracy exceeds all the above algorithms.

## 3 ARCHITECTURE

We propose a shallow feature extraction network with a large receptive field – SWNet – which consists of three parts: the primary feature extraction module (PFE), the atrous spatial pyramid pooling (ASPP) module and the feature fusion module (FFM). The network architecture is shown in Figure 1. The detailed parameters of the feature extraction network structure that is designed in this paper are shown in Appendix B.

The primary feature extraction network consists of three convolution layers with a kernel size of 3*3, each of which is followed by a batch normalization layer (Ioffe & Szegedy, 2015) and a ReLU layer. Only the stride of the first convolution layer is 2 to reduce the size of images. The other layers strides are set to 1 to retain more spatial structure information. Because of the shallow network structure, the size of the receptive field is limited. Therefore, inspired by Deeplab v2 (Chen et al., 2018), the ASPP module is added to the PFE module. In this module, dilated convolution layers with different dilation rate (e.g. 2, 4, 6, and 8) form four parallel branches. The four branches have receptive fields with different scales, which can complement each other. The outputs of four branches are added to obtain the feature maps containing multiscale information. Unlike the processing method of directly summing the feature maps with multiscale receptive fields, this paper adopts a feature fusion module to integrate the feature information of different scales. First, the feature maps that are obtained from each branch are concatenated to form a feature map group. Since the importance of the information that is contained in each feature map is different, inspired

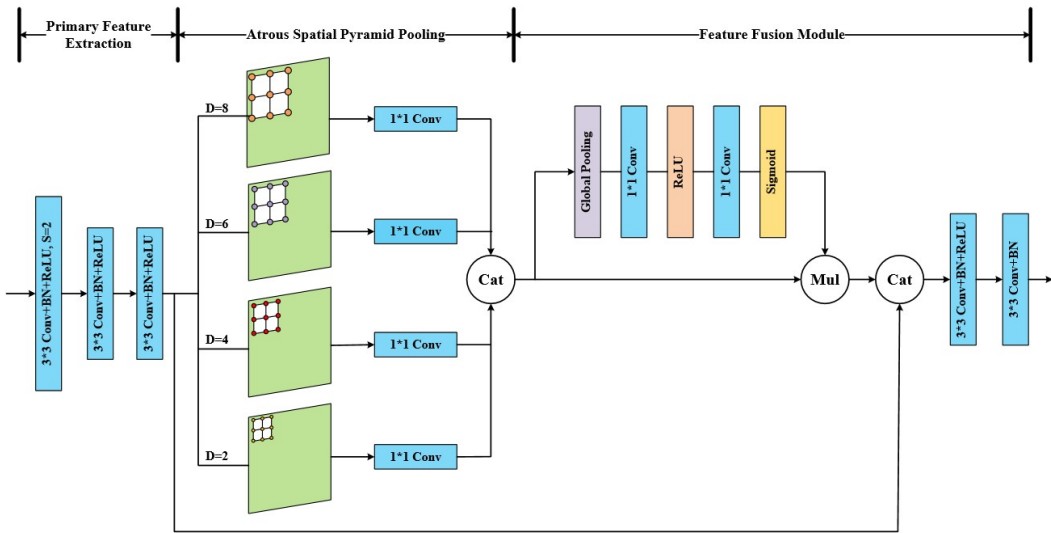

Figure 1: Feature extraction network architecture. D denotes the dilation rate, 1*1 conv denotes the convolution layer with kernel size of 1, Cat denotes concatenate operation, Mul denotes the channel-wise multiplication and BN denotes the batch normalization.

by SENet (Jie et al., 2017), this paper gives each feature map a specific weight. The feature fusion module is illustrated in Figure 1. The feature map group is converted into a 1D feature vector by global average pooling, a bottleneck structure is used to limit the number of parameters, and the weight of each channel is obtained by using a sigmoid function. The bottleneck structure is composed of two 1*1 convolution layers and a ReLU activation layer. The first convolution layer compresses the number of channels by $r$ times. After activation using the ReLU function, the number of channels is recovered by the second convolution layer. The weighted feature map group is obtained by multiplying the weight coefficient with the corresponding feature map. Then, the feature maps that are obtained by the PFE module are concatenated with the weighted feature map group through the skip connection, and the number of channels is compressed to 32 using two 3*3 convolution layers to obtain the final fusion feature maps.

## 4 EXPERIMENT

In this paper, we select PSMNet and GwcNet, two representative stereo matching algorithms, as our reference networks. The feature extraction network that is designed in this paper is used to replace the feature extraction part of the reference networks. The matching cost volume construction method adopts the most widely used shift and concatenation operation (the same as GC-Net and PSMNet). The 3D convolution, disparity regression and loss function use the same structure as the reference network. The network that is combined with PSMNet is called SWNet-P, and the network that is combined with GwcNet is called SWNet-G. In the following experiments, unless otherwise specified, the default network is SWNet-G.

In this section, we design experiments to study the effect of the depth of the feature extraction network, the size of the receptive field and the multiscale receptive fields on stereo matching. In section 4.1, we introduce the implementation details and the relevant information of the two datasets. In section 4.2, the shallow feature extraction network is compared with other deep networks to explore the effect of the network depth. In section 4.3, we calculate and test the size of the receptive field of the dilated convolution, and verify the effect of a large receptive field on stereo matching. In section 4.4, two important parameters of the ASPP module–the dilation rate and the number of branches–were tested to verify the effect of the fusion of multiscale receptive fields. In section 4.5, the stereo matching results that are generated by SWNet-P and SWNet-G are uploaded to KITTI, a third-party evaluation website, and compared with other advanced algorithms.

### 4.1 IMPLEMENTATION DETAILS AND DATASETS

We use Pytroch to implement the feature extraction network (SWNet) that was proposed by this paper. The whole model uses the Adam method for end-to-end training, where $\beta_1 = 0.9, \beta_2 = 0.99$. For all datasets, the training images are randomly cropped to a size of $512 \times 256$, and the intensity range of all pixels is normalized to [-1,1]. The maximum disparity is set to 192. For the SceneFlow dataset (Mayer et al., 2016), we conducted training for 10 epochs using a fixed learning rate of 0.001. For the KITTI 2015 dataset (Geiger, 2012), this paper used the model that was pretrained using SceneFlow data for fine-tune training. The model was trained for 300 epochs in total. For the first 200 epochs, the learning rate is set to 0.001, and for the later 100 epochs, the learning rate is adjusted to 0.0001. We trained the entire model on an NVIDIA 1080Ti GPU with the batch size set to 3. We take the end-point-error (EPE) of the SceneFlow test set and the three-pixel error (3-pix error) of the KITTI 2015 validation set as the evaluation bases.

This paper uses two open datasets for network training and testing.

**SceneFlow:** This dataset is a large-scale synthetic dataset containing 35454 training images and 4370 test images. The size of images is $960 \times 540$, and the dataset provides dense disparity maps as the ground truth. Those pixels whose disparity exceeds the maximum disparity set in this paper will be omitted when calculating the loss.

**KITTI 2015:** The dataset is a stereo dataset that is collected in a real street scene that contains 200 training images and 200 test images, the size of which is $1240 \times 376$. For the training subset, the sparse disparity map that is obtained by Li-DAR is provided as the ground truth. To facilitate the test, 40 pairs of stereo image in the training subset were randomly selected as the validation set, and the remaining 160 pairs of stereo image were selected as the training set.

### 4.2 THE EFFECT OF THE NETWORK DEPTH ON STEREO MATCHING

To explore the effect of the depth of the feature extraction network on the stereo matching accuracy, this paper modified the depth of the backbone of the feature extraction network of the reference networks and compared them with the feature extraction network that is designed in this paper.

Table 1: The effect of the network depth on stereo matching

| FEN | Cost Volume | 3D Conv | SceneFlow EPE | KITTI 3-pix error | Parameters |
|---|---|---|---|---|---|
| P+Res34 | c | P | 0.984 | 1.86% | 4.6 M |
| **P+Res50** | **c** | **P** | **0.963** | **1.81%** | **5.2 M** |
| SWNet(ours) | c | P | 0.872 | 1.74% | 2.3 M |
| G+Res34 | g&c | G | 0.911 | 1.69% | 6.3 M |
| **G+Res50** | **g&c** | **G** | **0.844** | **1.65%** | **6.9 M** |
| G+Res101 | g&c | G | 0.865 | 1.74% | 9.8 M |
| G+Res50 | c | G | 0.906 | 1.71% | 6.9 M |
| SWNet(ours) | c | G | 0.859 | 1.58% | 4.0 M |

Note: "FEN" means feature extraction network, "P" means the structure of PSMNet, "G" means the structure of GwcNet, "Res34,50,101" mean ResNet34,50,101 respectively, "c" means the matching cost volume constructed by concatenation and "g&c" means the matching cost volume constructed by group-wise correlation and concatenation. "Parameters" represents the number of parameters of the network. Bold text represents the default structure of the reference network. Due to the limitation of equipment performance, P+Res101 experiment cannot be carried out.

As seen from Table 1, when the backbone of the feature extraction network increases from 34 layers to 50 layers, the performance of the reference network on the SceneFlow dataset is significantly improved. End-point-error (EPE) decreases from 0.984 to 0.963 and from 0.911 to 0.844 in the SceneFlow dataset, respectively. However, as the network deepened and the backbone adopted the ResNet-101 structure, the stereo matching accuracy of GwcNet decreased. This result indicates that with the deepening of the network, the extracted feature information is more abstract and not suitable for the stereo matching task. Moreover, the large number of parameters in the deep

network makes the model training more difficult. When other parts of the network are the same, the matching accuracy of SWNet is close to or even better than the default structure of the reference network (with ResNet-50 as the backbone) in both datasets, and the number of parameters is greatly reduced. This result shows that simply increasing the network depth cannot improve the stereo matching accuracy. The shallow feature extraction network can extract and retain more details of the spatial structure, which is more suitable for the stereo matching task. In addition, the network has fewer parameters, lower training difficulty and a stronger generalization ability.

## 4.3 THE EFFECT OF THE SIZE OF THE RECEPTIVE FIELD ON STEREO MATCHING

The dilated convolution can enlarge the receptive field and solve the problem of a limited receptive field in a shallow network. However, a dilated convolution may cause some input neurons to fail, leading to a cavity in the receptive field. In this section, we propose the concepts of the theoretical receptive field (TRF) and the effective receptive field (ERF). The theoretical receptive field refers to the region that can be observed in the input space for a neuron in the convolutional neural network. The effective receptive field refers to the set of input neurons that are actually connected to a higher level neuron, excluding the invalid neurons in the receptive field. In this paper, the mathematical calculation methods (Yu & Koltun, 2016) of the two kinds of receptive fields are given (the specific derivation process is shown in the Appendix A), and a simple experiment is designed to intuitively demonstrate the effect of the dilated convolution on the receptive field by means of visualization (Luo et al., 2017).

### 4.3.1 MATHEMATICAL CALCULATION METHOD

The size of theoretical receptive field is calculated as follows:

$$r_n = r_{n-1} + (k_n - 1) d_n \prod_{i=1}^{n-1} s_i \tag{1}$$

$r_n$ denotes the size of the theoretical receptive field corresponding to each neuron in the $n^{th}$ layer, $k_n$ denotes the kernel size, $d_n$ denotes the dilation rate and $s_i$ denotes the stride of the $i^{th}$ convolution layer. The size of effective receptive field is calculated as follows:

$$r'_n = r_n - p_0 (k_n - 1) \tag{2}$$

$p_0$ denotes the number of the invalid neurons of the input layer, and the calculation method is shown as follows:

$$\begin{cases} p_n = d_{n+1} - 1 \\ p_{n-1} = p_n (N_n + 1) - M_n + 1 \end{cases} \tag{3}$$

in which

$$N_n = max(2s_n - k_n, 0) \tag{4}$$

$$M_n = k_n - s_n + 1 \tag{5}$$

To describe the relationship between the theoretical receptive field and the effective receptive field, we proposed the concept of the density of the receptive field and the calculation method is shown as follows.

$$Q = \left(\frac{r'_n}{r_n}\right)^2 \times 100\% \tag{6}$$

According to the above formulas, taking the primary feature extraction network that is designed in this paper as an example, the corresponding relationship between the dilation rate and the size of receptive field is shown in Table 2.

As seen from Table 2, the size of the theoretical receptive field continues to grow as the dilation rate increases. When the dilation rate is 12, the size of the theoretical receptive field is close to 16 times that of the ordinary convolution (the dilation rate is 1). Limited by the number of network layers, the size of the ERF increases to 33*33 and then does not change, and the density of the receptive field rapidly decreases.

Table 2: The effect of the dilation rate on the size and density of the receptive field

| Dilation Rate | 1 | 2 | 4 | 6 | 8 | 10 | 12 |
|---|---|---|---|---|---|---|---|
| the size of TRF $r'_n$ | 15*15 | 19*19 | 27*27 | 35*35 | 43*43 | 51*51 | 59*59 |
| the size of ERF $r_n$ | 15*15 | 19*19 | 27*27 | 33*33 | 33*33 | 33*33 | 33*33 |
| the density of RF $Q$ | 100% | 100% | 100% | 89% | 59% | 42% | 31% |

### 4.3.2 VISUALIZATION EXPERIMENT METHOD

To verify the results of the above mathematical derivation, a intuitive experiment is designed in this section. The effect of the dilation rate on the size of the receptive field is visually demonstrated.
If the neurons in the low-level network are regarded as receptors in the human nervous system, the neurons in the high-level network represent the higher nerve center, and each nerve center is connected with multiple receptors in the lower layer. Therefore, we can determine whether this receptor is related to the nerve center by examining the response of the higher nerve center after giving certain stimuli to the lower receptor. Specifically, this paper applies external stimuli to each pixel of the input image in turn (such as increasing the RGB value by 10) to detect the value change of the high-level neuron. If the value changes, the input neuron is related to the high-level neuron. The more obvious the change is, the stronger the correlation is.

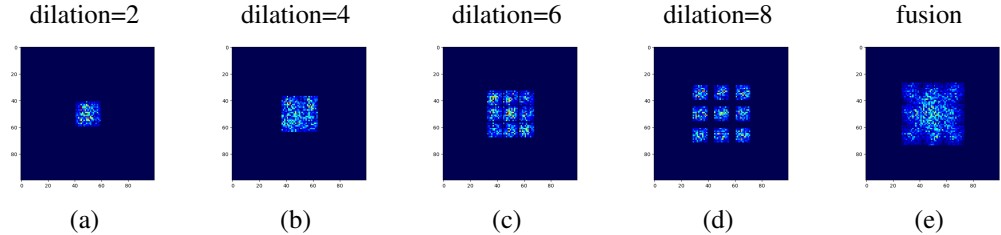

Figure 2: The effect of the dilation rate on the size and density of the receptive field. We normalized the value change to the range of [0,1] and mapped it to the input images. For clarity, only the 100*100 pixel images that are centered on the high-level neuron are retained.

As shown in Figure 2, the color part in the figure indicates the corresponding receptive field of the high-level neuron. The brighter the color is, the stronger the correlation is between the input neuron and the central neuron. It is obvious that the size of the receptive field is increasing with the dilation rate. However, when the dilation rate is greater than 6, there are cavities in the receptive field, and with the continuous increase of the dilation rate, the sizes of cavities rapidly grow, which is consistent with our theoretical derivation. To solve the problem of cavities in the receptive field, the ASPP module and feature fusion module are used to fuse the convolution layers with different dilation rates. By this means, the cavities in the receptive field are effectively compensated while maintaining a large receptive field, as shown in Figure 2(e).
To verify the effect of the size of receptive field and density on stereo matching, this paper conducted ablation experiments on the ASPP module and feature fusion module. The experiment results are shown in Table 3.

Table 3: The effect of the size of receptive field and density on stereo matching

| PFE | ASPP | FFM | SceneFlow EPE | KITTI 3-pix error |
|---|---|---|---|---|
| ✓ | | | 0.931 | 1.81% |
| ✓ | ✓ | | 0.879 | 1.71% |
| ✓ | ✓ | ✓ | 0.859 | 1.58% |

As seen in Table 3, the matching error of the primary feature extraction network is high, because its structure is too shallow and the size of receptive field is limited; therefore, more context information cannot be extracted. The introduction of the dilated convolution and ASPP module effectively enlarge the receptive field, and improve the stereo matching accuracy. The EPE of SceneFlow dataset decreases from 0.931 to 0.879. The feature fusion module can better fuse the information of the multiscale receptive fields and further reduce the EPE to 0.859. The experiments of this section shows that the dilated convolution can effectively enlarge the theoretical receptive field, but there will be cavities in the receptive field, leading to the partial loss of information. The ASPP module and feature fusion module can fuse the feature maps with multiscale receptive field and solve the information loss that is caused by the dilated convolution. This structure can obtain a large receptive field, and also ensure that the receptive field is dense enough to provide more context information and improve the matching accuracy of ill-posed regions.

### 4.3.3 THE EFFECT OF MULTISCALE RECEPTIVE FIELDS ON STEREO MATCHING

The ASPP module can enlarge the receptive field and provide information on the multiscale receptive fields. In this section, a series of experiments are designed for the two hyper-parameters of the ASPP module: the dilation rate and the number of branches. For the dilation rate, we carried out experiments on two groups of dilation rate parameters with base 2 and base 3. With respect to the number of branches, the ASPP module with 4 and 8 branches were tested. The experimental results are shown in the Table 4.

Table 4: The effect of the size of receptive field and density on stereo matching

| ASPP module | SceneFlow EPE | KITTI 3-pix error | parameters |
|---|---|---|---|
| [2,4,6,8] | 0.859 | 1.58% | 4.0 M |
| [3,6,9,12] | 0.851 | 1.70% | 4.0 M |
| [2,4,6,8,10,12,14,16] | 0.864 | 1.68% | 4.4 M |
| [3,6,9,12,15,18,21,24] | 0.842 | 1.62% | 4.4M |

As seen from Table 4, the stereo matching accuracy improved as the number of branches increased. The EPE decreased from 0.851 to 0.842 (base 3) in the SceneFlow dataset, and the 3-pixel error decreased from 1.70% to 1.62% in the KITTI 2015 dataset, while the number of parameters will increase from 4.0M to 4.4M. The dilation rate with the base of 3 is generally better than that with the base of 2, and the EPE decreases by approximately 2% on average. However, in the KITTI 2015 dataset, ASPP module with dilation rate of [2, 4, 6, 8] got the best performance, with 3-pixel error is 1.58%. This result indicates that more receptive fields with different scales can be obtained by adding branches of the ASPP module. A small scale receptive field can extract local detailed structural information, and a large scale receptive field can obtain more context information. The feature extraction network should extract as much information of different scales as possible to improve the overall matching accuracy.

### 4.3.4 KITTI 2015 BENCHMARK

We uploaded the results that were generated by SWNet to KITTI and compared these results with those of other excellent stereo matching algorithms. The KITTI 2015 leaderboard is shown in the Table 5.

As seen from Table 5, SWNet has the lowest matching error compared with existing stereo matching algorithms. Compared with the PSMNet and GwcNet reference networks, the error rate was reduced by 3.4% and 1.9%, respectively, and the number of network parameters was decreased by 56% and 42%, respectively. This result shows that the shallow feature extraction network with a large receptive field can better extract and retain the feature information that is needed for the stereo matching task and improve the stereo matching accuracy. At the same time, the shallow feature extraction network can reduce the number of network parameters and the network training and deployment difficulties. In terms of the processing speed, the network performance is related to the performance of the computing platform. Since we only use a common GPU for calculations,

Table 5: The KITTI 2015 leaderboard. "D1" represents the percentage of stereo disparity outliers. "bg" represents the background region, "fg" represents the foreground region, and "all" represents the entire region. "Runtime" represents the time to process a pair of stereo images. The bold text represents the improved stereo matching algorithm in this paper.

| Network | All Pixels | | | Runtime | Parameters |
|---|---|---|---|---|---|
| | D1-bg | D1-fg | D1-all | | |
| GC-Net | 2.21% | 6.16% | 2.87% | 0.9 s | 3.5 M |
| PSMNet | 1.86% | 4.62% | 2.32% | 0.45 s | 5.2 M |
| CFP-Net | 1.90% | 4.39% | 2.31% | 0.9 s | - |
| Stereo-DRNet | 1.72% | 4.95% | 2.26% | 0.23 s | - |
| **SWNet-P(ours)** | **1.81%** | **4.41%** | **2.24%** | **0.4 s** | **2.3 M** |
| GwcNet | 1.74% | 3.93% | 2.11% | 0.32 s | 6.9M |
| MUCA | 1.66% | 4.27% | 2.09% | 0.9 s | - |
| **SWNet-G(ours)** | **1.68%** | **4.02%** | **2.07%** | **0.6 s** | **4.0 M** |

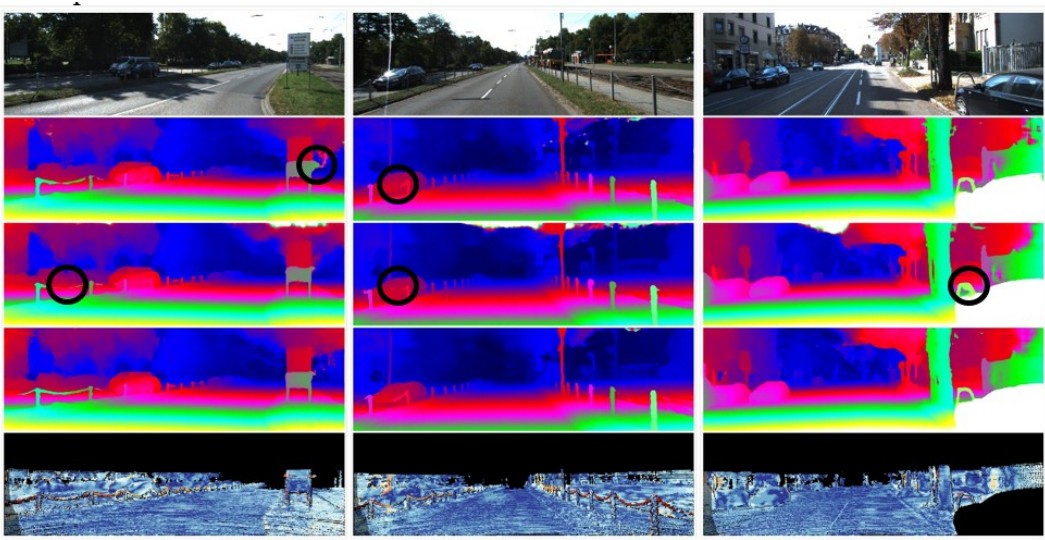

Figure 3: The disparity maps and error maps of the KITTI 2015 dataset. From top to bottom are the left image of the input; the disparity maps of the PSMNet, GwcNet and SWNet; and the error map of SWNet.

the processing speed is slightly inferior to the performance of the reference network.

As shown in Figure 3, compared with the PSMNet and GwcNet, the SWNet retains more detailed structural information, so it has a better matching effect in areas such as iron chains, traffic signs and railings (the areas that are marked by black circles in the figure). In addition, due to the use of the ASPP module to enlarge the receptive field, the SWNet still maintains a high matching accuracy on large-scale objects such as vehicles, buildings, trees and so on.

## 5 CONCLUSION

Focusing on the feature extraction part of a stereo matching network, this paper proposes a novel network structure, which abandons the popular deep convolution neural network and use the shallow network structure to extract and retain more basic feature information. To solve the problem that the receptive field of a shallow network is limited, this paper introduces the ASPP module and obtains multiscale receptive fields by adding convolution branches with different dilation rates. By using the feature fusion module, the feature maps with multiscale receptive fields are fused together to solve the information loss problem that is caused by dilated convolution. Finally, a large and

dense receptive field is obtained. The shallow feature extraction network with a large receptive field can provide more suitable feature information for stereo matching task, with fewer parameters and lower training difficulty. Using the SWNet to replace the feature extraction part of the existing network can effectively improve the stereo matching accuracy.

## ACKNOWLEDGEMENTS

This work was supported by the National Natural Science Foundation of China (51975434), the 111 Project(B17034)and the Excellent Dissertation Cultivation Funds of Wuhan University of Technology (2018-YS-033).

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

## A  APPENDIX

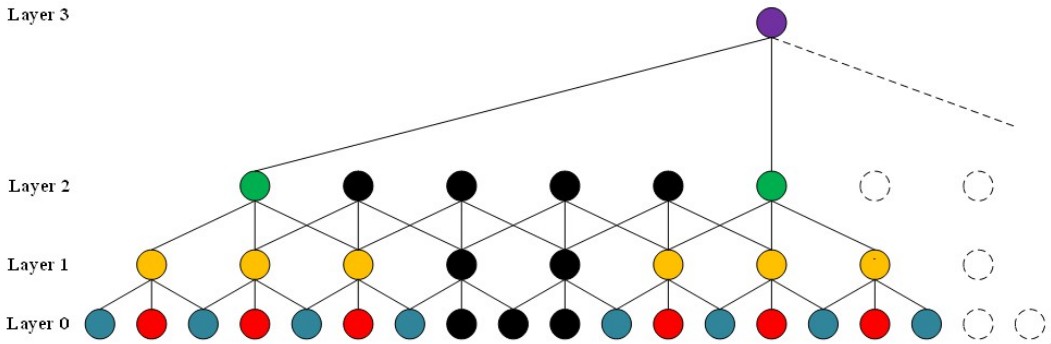

Figure 4: Schematic diagram of neurons corresponding to receptive fields.

To clearly explain the calculation process of the theoretical receptive field and effective receptive field, the 2D convolution neural network is simplified into a 1D neural network similar to multilayer perceptron (MLP). The connection relationship between its neurons is shown in Figure 4, where each circle represents one neuron. Limited by the size of the image, only half of the receptive field of the neuron is shown. The receptive field of the neuron in layer 0 (input layer) is 1, that is $r_0 = 1$. The receptive field of the neuron in layer 1 is $r_1 = r_0 \times k_1 = 1 \times 3 = 3$. The receptive field of neurons in layer 2 is $r_2 = r_1 \times k_2 = 3 \times 3 = 9$ , but since neurons are not independent of each other, there are overlaps between their receptive fields, so the overlaps must be subtracted when calculating the size of the receptive field. The number of neurons in the overlapping part is related to the kernel size and the convolution stride. As shown in Figure 4, the kernel size of the neurons in layer 2 is three. Then there are two overlaps in the corresponding receptive field, and the number of neurons that is contained in each overlaps is one. Therefore, the number of neurons that is contained in all overlaps is as follows.

$$(k_2 - 1)(r_1 - s_1) = 2 \times 1 = 2 \tag{7}$$

Then the size of receptive field of neuron in layer 2 should be modified as

$$r_2 = r_1 \times k_2 - (k_2 - 1)(r_1 - s_1) = 3 \times 3 - 2 \times 1 = 7 \tag{8}$$

It is worth noting that, in the convolution neural network, as the number of convolution layers increases, the impact of convolution stride is cumulative. Therefore, the size of the receptive field of the neuron in layer *n* should be formulated as

$$r_n = r_{n-1} \times k_n - (k_n - 1)(r_{n-1} - \prod_{i=1}^{n-1} s_i) = r_{n-1} + (k_{n-1}) \prod_{i=1}^{n-1} s_i \tag{9}$$

For dilated convolution, the kernel size should be modified as

$$k'_n = k_n + (k_n - 1)(d_n - 1) \tag{10}$$

By substituting formula (10) into formula (9), the size of the theoretical receptive field of the dilated convolution can be calculated as

$$r_n = r_{n-1} + (k'_n - 1) \prod_{i=1}^{n-1} s_i$$

$$= r_{n-1} + [k_n + (k_n - 1)(d_n - 1) - 1] \prod_{i=1}^{n-1} s_i \tag{11}$$

$$= r_{n-1} + (k_n - 1)d_n \prod_{i=1}^{n-1} s_i$$

For the size of the effective receptive field, this paper only studies the case when the convolution stride is smaller than the kernel size, which is $k_n > s_n$. As shown in Figure 4, the kernel of the neuron in layer 3 is dilated, and the information of some low-level neurons will not be transmitted to the neuron in layer 3, which are called invalid neurons (black circles in Figure 4). The maximum number of continuous invalid neurons in layer 2 is the dilation rate of layer 3 minus 1, which is $p_2 = d_3 - 1 = 5 - 1 = 4$. The maximum number of continuously invalid neurons in layer 0-1 is related to the connection relationship between network layers. To describe this relationship, this paper introduces the concepts of exclusive subneurons and shared subneurons. Subneurons refer to the low-level neurons that are directly connected to the neurons in higher layers. As shown in Figure 4, the green neurons are the subneurons of purple neurons, while the black neurons are not. An exclusive subneuron refers to the only sub-neuron in layer *(n-1)* that is connected to a neuron in layer *n*. As shown in Figure 4, the red neurons are the exclusive subneurons of the yellow neurons. Under the 1D condition, each neuron has two adjacent neurons, and there is overlap between the subneurons of every two neurons. Therefore, the number of exclusive subneurons of a neuron in layer *n* can be calculated as

$$N_n = k_n - (k_n - s_n) \times 2 = 2s_n - k_n \tag{12}$$

However, the number of exclusive subneurons should be non-negative, with a minimum value of 0. Therefore, a non-negative constraint is added to formula (12)

$$N_n = max(2s_n - k_n, 0) \tag{13}$$

Therefore, if one neuron in layer *n* fails, it will directly lead to the failure of $N_n$ subneurons in layer *(n-1)*.

A shared subneuron refers to the subneuron that is connected with multiple neurons in higher layers. As shown in Figure 4, the blue neurons are the shared neurons of the yellow neurons. A shared subneuron in layer *(n-1)* is connected to $M_n$ neurons in layer *n*. In other words, if there are $M_n$ continuously invalid neurons in layer *n*, there will be one invalid neuron in layer *(n-1)*. The calculation method of $M_n$ is

$$M_n = k_n - s_n + 1 \tag{14}$$

Comprehensively considering the exclusive subneurons and shared subneurons, when there are $p_n$ invalid neurons in layer *n*, the number of invalid neurons in layer *(n-1)* is

$$p_{n-1} = p_n N_n + (p_n - M_n + 1) = p_n(N_n + 1) - M_n + 1 \tag{15}$$

If the invalid neuron in layer *n* is directly caused by the dilated convolution, the number of invalid neurons in layer *n* is

$$p_n = d_{n+1} - 1 \tag{16}$$

As shown in Figure 4, the number of invalid neurons in layer 2 is $p_2 = d_3 - 1 = 5 - 1 = 4$. The numbers of invalid neurons in layer 1 and 0 are $p_1 = 4 \times (0 + 1) - 3 + 1 = 2$ and $p_0 = 2 \times (1 + 1) - 2 + 1 = 3$, respectively.

The size of the effective receptive field should be the size of theoretical receptive field minus the number of invalid neurons in layer 0. The calculation method is shown in formula (17)

$$r'_n = r_n - p_0(k_n - 1) \tag{17}$$

## B    APPENDIX

K denotes the convolution kernel size, C denotes the number of output channels, S denotes the convolution stride, D denotes the dilation rate, BN denotes the batch normalization layer, ReLU denotes the activation layer, H denotes the height of the image and W denotes the width of the image. Concat stands for the concatenation operation of feature maps, and SElayer stands for assigning weights to each feature map.

Table 6: The network structure parameter of SWNet

| Input | Setting | Output | The input size | The output size |
|-------|---------|--------|----------------|-----------------|
| primary feature extraction module | | | | |
| left/right | | input | H*W*3 | H*W*3 |
| input | K=3*3,C=32,S=2, D=1,BN,ReLU | conv_0 | H*W*3 | 1/2H*1/2W*32 |
| conv_0 | K=3*3,C=64,S=1, D=1,BN,ReLU | conv_1 | H*W*32 | 1/2H*1/2W*64 |
| conv_1 | K=3*3,C=128,S=1, D=1,BN,ReLU | conv_2 | H*W*64 | 1/2H*1/2W*128 |
| ASPP module | | | | |
| conv_2 | K=3*3,C=32,S=1, D=2,BN,ReLU | branch_1_s | 1/2H*1/2W*128 | 1/2H*1/2W*32 |
| branch_1_s | K=3*3,C=32,S=1, D=1,BN,ReLU | branch_1 | 1/2H*1/2W*32 | 1/2H*1/2W*32 |
| conv_2 | K=3*3,C=32,S=1, D=4,BN,ReLU | branch_2_s | 1/2H*1/2W*128 | 1/2H*1/2W*32 |
| branch_2_s | K=3*3,C=32,S=1, D=1,BN,ReLU | branch_2 | 1/2H*1/2W*32 | 1/2H*1/2W*32 |
| conv_2 | K=3*3,C=32,S=1, D=6,BN,ReLU | branch_3_s | 1/2H*1/2W*128 | 1/2H*1/2W*32 |
| branch_3_s | K=3*3,C=32,S=1, D=1,BN,ReLU | branch_3 | 1/2H*1/2W*32 | 1/2H*1/2W*32 |
| conv_2 | K=3*3,C=32,S=1, D=8,BN,ReLU | branch_4_s | 1/2H*1/2W*128 | 1/2H*1/2W*32 |
| branch_4_s | K=3*3,C=32,S=1, D=1,BN,ReLU | branch_4 | 1/2H*1/2W*32 | 1/2H*1/2W*32 |
| feature fusion module | | | | |
| branch1~4 | Concat | cat | 1/2H*1/2W *(32*4) | 1/2H*1/2W*128 |
| cat | SElayer | se | 1/2H*1/2W*128 | 1/2H*1/2W*128 |
| se | K=3*3,C=128,S=1, D=1,BN,ReLU | fusion | 1/2H*1/2W*128 | 1/2H*1/2W*128 |
| fusion,conv_2 | K=3*3,C=128,S=2, D=1,BN,ReLU | conv_3 | 1/2H*1/2W *(128+32) | 1/4H*1/4W*128 |
| conv_3 | K=3*3,C=32,S=1, D=1,BN,ReLU | conv_4 | 1/4H*1/4W*128 | 1/4H*1/4W*32 |

