# OpenReview forum: "A shallow feature extraction network with a large receptive field for stereo matching tasks"
_ICLR.cc/2020/Conference — Reject_

### Official Review · AnonReviewer1 · 2019-10-23
**Official Blind Review #1**

**Rating:** 1

**Review:**

The paper at hand argues that shallow feature extraction networks should be favored for the computer vision task of stereo matching, rather than the commonly used deep ResNet backbones. To that end, a model is proposed consisting of three convolutional feature extraction layers only. As this makes it impossible to capture global context, dilated convolutions with varying dilations are applied to the extracted features in parallel and concatenated. Finally, a fusion module implemented via channel attention is applied.

I found this paper lacking in terms of contributions. While the motivation for retaining detailed feature information makes sense for the task in question, it is not clear why it requires replacing the ResNet feature extraction with a very shallow network; an alternative would be to add skip-connections originating from lower levels. Both dilated convolutions and the fusion module are not novel. The paper addresses the need for a fusion module in great detail. However, the insight that a large dilation on top of features computed from a limited receptive field will result in a non-continuous receptive field is a rather trivial one. Hence I don't see the need for section 4.3.1 and 4.3.2, as well as the somewhat complicated deduction in appendix A. The individual modules are ablated, but what about the number of layers for feature extraction? Why settle for 3 rather than 1, 2, or 4?

The paper does compare against other methods for stereo matching, but I was wondering why the KTTI leader-board excerpt does not include the better results from http://www.cvlibs.net/datasets/kitti/eval_scene_flow.php?benchmark=stereo ? The result tables are also hard to parse as bold numbers do not correspond to best performance. The conclusion claims that the proposed shallow architecture exhibits "lower training difficulty" but I did not see any support for this in the experiments.

Overall, I think that this paper should be rejected on the basis of insufficient contributions.

**Experience Assessment:**

I have read many papers in this area.

**Review Assessment: Checking Correctness Of Derivations And Theory:**

I assessed the sensibility of the derivations and theory.

**Review Assessment: Checking Correctness Of Experiments:**

I assessed the sensibility of the experiments.

**Review Assessment: Thoroughness In Paper Reading:**

I read the paper at least twice and used my best judgement in assessing the paper.

---

### Official Review · AnonReviewer3 · 2019-10-24
**Official Blind Review #3**

**Rating:** 3

**Review:**

This paper presents an algorithm for stereo image matching that attempts to capture improved representations of detailed spatial structure information, in particular, by increasing the size of the receptive field.  The paper shows that this leads to a major reduction in the number of model parameters (42% in one case) with comparable performance on the KITTI2015 data set.

I like the driving principle of the authors' approach (that stereo image matching relies more heavily on low-level features, and that higher level "semantic" features are not as critical) compelling.  I would have really like to have seen the authors do some analysis of the features that they do extract, so that the reader can get a deeper insight into why their method works. The paper could be improved by providing more this kind of analysis and by adding more motivate for why low-level features are more important for stereo matching.

I'm concerned that the paper only present results on one, small (200+200 images) data set.  The paper would be much stronger if the authors tested on more, and varied data sets.

Is it simply a network complexity issues or is there something else?

The related work section appears to be just a laundry list of methods.  The paper would be stronger if the authors provided more interpretation of the strengths and weaknesses of these methods, some insight into why they work, and why the proposed method is better.

The authors' method claims to use a 1x1 convolution layer.  Is that correct?  Sounds like simple multiplication.  Explain what it different.

The authors' reporting of their results appears muddled.  They claim the error rate was "reduced 3.4% and 1.9%" in Table 5.  I could not figure out which numbers they were talking about.  In most cases, the authors' method was not the best.

Minor point:  The authors say "conclusion" when I think they mean "oclusion".

**Experience Assessment:**

I have read many papers in this area.

**Review Assessment: Checking Correctness Of Derivations And Theory:**

I assessed the sensibility of the derivations and theory.

**Review Assessment: Checking Correctness Of Experiments:**

I assessed the sensibility of the experiments.

**Review Assessment: Thoroughness In Paper Reading:**

I read the paper at least twice and used my best judgement in assessing the paper.

---

### Official Review · AnonReviewer4 · 2019-10-27
**Official Blind Review #4**

**Rating:** 6

**Review:**

This is a paper about a rather specialized area of computer vision (stereo matching),
and it's not really a theoretical paper; it's about an improved network topology for
a very specific task.
My feeling is that this belongs in a computer vision conference, where people would be better
able to appreciate it.
It does relate specifically to learning representations, though, so perhaps it has a chance in ICLR?
I'm putting "weak accept" but take this as with weak confidence as I am not really a computer vision person.

**Experience Assessment:**

I do not know much about this area.

**Review Assessment: Checking Correctness Of Derivations And Theory:**

I did not assess the derivations or theory.

**Review Assessment: Checking Correctness Of Experiments:**

I did not assess the experiments.

**Review Assessment: Thoroughness In Paper Reading:**

I made a quick assessment of this paper.

---

### Decision · Program_Chairs · 2019-12-19

**Decision:**

Reject

**Comment:**

The paper proposed the use of a shallow layers with large receptive fields for feature extraction to be used in stereo matching tasks. It showed on the KITTI2015 dataset this method leads to large model size reducetion while maintaining a comparable performance.

The main conern on this paper is the lack of technical contributions:
* The task of stereo matching is very specialized one, simply presenting the model size reduction and performance is not interesting to general readers. Adding more analysis that help understanding why the proposed method helps in this particular task and for what kind of tasks a shallow feature instead a deeper one is perferred. In that way, the paper would be addressing much wider audiences.
* The discussions on related work is not thorough enough, lacking of analysis of pros and cons between different methods.